# TWINKLE: An open-source two-photon microscope for teaching and research

Manuel Schottdorf[1,2]*, P. Dylan Rich[1], E. Mika Diamanti[1], Albert Lin[1,3], Sina Tafazoli[1], Edward H. Nieh[1,4], Stephan Y. Thiberge[1,5]*

**1** Princeton Neuroscience Institute, Princeton University, Princeton, NJ, United States of America, **2** Psychological and Brain Sciences, University of Delaware, Newark, DE, United States of America, **3** Center for the Physics of Biological Function, Princeton University, Princeton, NJ, United States of America, **4** Department of Pharmacology, School of Medicine, University of Virginia, Charlottesville, VA, United States of America, **5** Bezos Center for Neural Circuit Dynamics, Princeton University, Princeton, NJ, United States of America

* maschott@udel.edu (MS); thiberge@princeton.edu (SYT)

## Abstract

Many laboratories use two-photon microscopy through commercial suppliers, or homemade designs of considerable complexity. The integrated nature of these systems complicates customization, troubleshooting, and training on the principles of two-photon microscopy. Here, we present "Twinkle": a microscope for Two-photon Imaging in Neuroscience, and Kit for Learning and Education. It is a fully open, high performing and easy-to-set-up microscope that can effectively be used for both education and research. The instrument features a >1 mm field of view, using a modern objective with 3 mm working distance and 2 inch diameter optics combined with GaAsP photomultiplier tubes to maximize the fluorescence signal. We document our experiences using this system as a teaching tool in several two week long workshops, exemplify scientific use cases, and conclude with a broader note on the place of our work in the growing space of open scientific instrumentation.

## Introduction

Two-photon microscopy [1, 2] is a key technology across the modern life sciences [3], for example, in physiology [4, 5], cancer research [6], plant biology [7], and neuroscience [8, 9]. New scientific use cases and technological improvements are produced at a remarkable rate [10]. However, these instruments are often published without detailed building instructions and explanations. This suggests the need for an easy-to-set-up microscope, at reasonable cost, that can effectively be used for education, dissemination, methods development and research. We developed a high-performance and cost-effective two-photon microscope that can easily be produced in many neuroscience laboratories. Reflecting its use for "**TW**o-photon **I**maging in **N**euroscience, and **K**it for **L**earning and **E**ducation", we chose the acronym Twinkle. In this article, we share our design, document the performance of the system, explore possible research applications, and report our experiences collected during several teaching workshops. Complete CAD drawings, bill of materials, optics simulations, and detailed building instructions are provided in the supplement.

**Data Availability Statement:** All information, including CAD files, electronics design, and optics design are available on github: https://github.com/BrainCOGS/Microscope.

**Funding:** MS and MD are supported by NIH grant U19NS132720 (https://www.nih.gov/). MS is also supported by a C.V. Starr fellowship and a Burroughs Wellcome Fund's Career Award at the Scientific Interface (https://www.bwfund.org/). AL was supported by the NSF (https://www.nsf.gov/) through the Center for the Physics of Biological Function (PHY-1734030). The funders did not play any role in the study design, data collection, analysis, decision to publish, or preparation of the manuscript.

## Materials and methods

In two-photon laser-scanning microscopy [1], laser light is focused to a small excitation volume, the "focal point", which is moved across the sample by the intermediary of resonant and galvanometric scanning mirrors. As the focal point probes different locations, different quantities of fluorescence light are produced by the sample. This light is collected by a sensitive detector in the form of an electronically measurable photocurrent. An image is then formed, mapping the detector photocurrent values to the locations of excitation volume: low and high photocurrents, respectively, becoming dark and bright pixels. Central to two-photon fluorescence excitation is the light source: a femtosecond-pulsed infrared laser of sufficient pulse energy to achieve, when focused by an objective lens, a photon density high enough for simultaneous absorption of two photons by a fluorescent molecule. Here, we design an open and high-performing microscope that uses these principles. As such, we have made all optical and mechanical designs, electronics, bill of materials (BOM), CAD assembly, testing results, and relevant schematics open to the public. Excluding laser and optical table, the cost is around US-$ 110k (2024). Including laser and table, the cost is around US-$ 200k (2024). A narrative of the design and the high-level view is presented here. For further details, we refer the reader to S1 Appendix. This contains illustrated construction instructions and technical details. All code, figures and text, CAD designs, optics simulations, and BOM are available on https://github.com/BrainCOGS/Microscope and on Zenodo [11].

### Design specifications

We aimed for a mechanical and optical assembly using as many off-the-shelf components as possible, with only few custom aluminium parts that can be machined in any university machine shop. Our design facilitates the adaptation for *in vivo* imaging by providing a large space for the organism and ancillary hardware. Our system is made cost-effective, in part, by the availability of femtosecond laser systems based on fiber technology. Operating at a fixed wavelength, these systems come at the fraction of the cost of a tunable Ti:Sapphire laser [12].

More specifically, we aimed for (1) $\gtrsim$ 20 cm of free space around the objective in all directions to aid integration of the microscope with various peripheral equipment such as behavior boxes or a sample stage, and (2) imaging on a $\approx 700 \times 700 \ \mu m^2$ minimum size field of view which is typical for imaging brain tissue at cellular resolution. (3) Compared to earliest open designs [13, 14], our system can make use of the large Numerical Aperture (NA) of modern long-working-distance objectives to collect more fluorescence, but requires to adapt our design to their larger back apertures [15].

An overview of the microscope is shown in Fig 1. Fig 1A and 1B shows the system as a cartoon. Full details will be provided below. In short, we chose a 920 nm femtosecond pulsed laser, whose beam diameter is extended by a 2× magnification telescope, before it reaches the scanning mirrors. The scanning mirrors are purchased as a mount-assembled set. Located as close as possible to each other, as we will see in the alignment procedure, neither of them can be exactly positioned in the focal plane of the scan lens. The choice of mounted scanners simplifies the design and construction, but leads to optical distortions across the field of view. We found that this distortion does not exceed 5% in our design (see below). The combined scan and tube lenses magnify the beam further to fill the objective's back aperture. Fluorescence from the specimen is then collected and brought to the aperture of two GaAsP photomultiplier tubes with suitable color filters. Fig 1C shows the CAD design of the entire assembly with important parts of the optical path labeled. Fig 1D shows an annotated picture of the microscope head. Notice that compared to the cartoon in Fig 1A and 1B, the optical path is folded with several mirrors. These mirrors make the microscope assembly more compact.

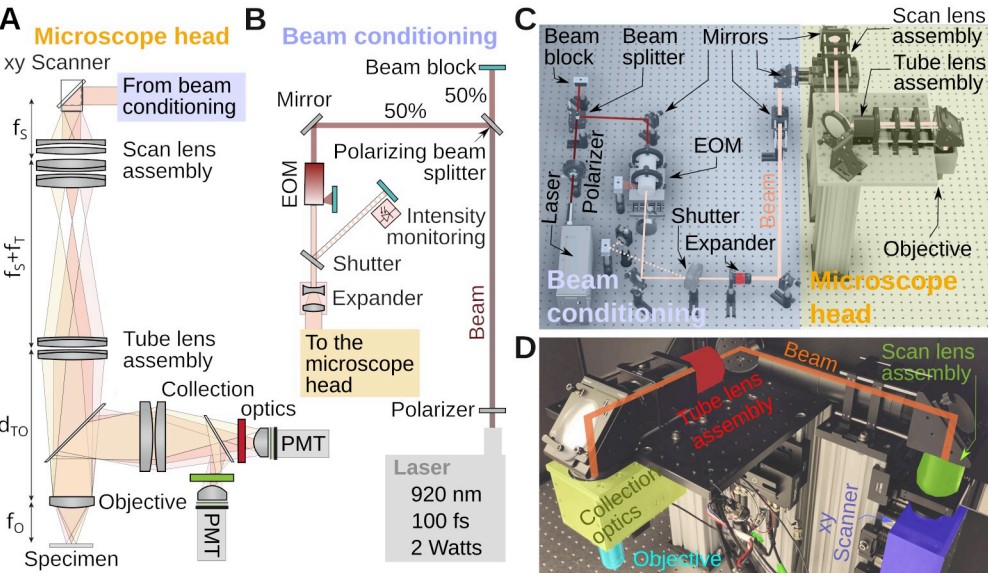

**Fig 1. System overview. A)** Key components of the microscope head. Shaded colors illustrate beam deflection angles. **B)** Cartoon of the beam conditioning components. First the beam is split by 50%, the intensity adjusted with an electro-optic modulator (EOM) and the beam width with a telescope before entering the microscope head. **C)** 3D drawing of the full microscope, highlighting beam conditioning subsystem (blue) and the microscope head (yellow). For scale, the spacing between two holes on the optical table is 1 inch or 2.5 cm. **D)** Photograph of the microscope head with key elements labeled. This section is visible in panel A.

## Assembly on the table

Various components are used between the laser and the microscope head to condition the beam, and for intensity control, Fig 1B. Out of the laser head, the beam first encounters a waveplate (AHWP10M-980, Thorlabs). This is a birefringent crystal that can rotate the linear polarization of the laser to any angle. Next, the beam travels through a polarizing beamsplitter cube (PBS103, Thorlabs). This splits the beam into two orthogonal polarization components whose relative intensity is adjusted by the orientation of the waveplate. We set the orientation of the waveplate to split the beam into equal power components, the second one being eventually used for a second microscope installed on the other extremity of the table. In the cartoon, this second beam is sent to a beam block (LB1, Thorlabs). We want to stress that this beam-splitting hardware is optional. However, modern lasers have enough power to supply two microscopes at the same time. It is therefore often strategic to split the beam into two beams, with half of the intensity each, to allow one laser (the most expensive component) to power two systems. Next, the beam passes through an electro-optic modulator (EOM), a Pockels cell (350–80-LA-02, Conoptics). This device allows rapid and electronic control of the laser's power level. It uses a crystal whose refractive index is controlled by an external electric field, combined with a polarizer. This can be thought of as a voltage-controlled wave plate, in which the electric field controls how much light travels through the polarizer. Next, the beam travels through an open mechanical 6 mm diameter shutter (LS6S2ZM1, Vincent Associates). When closed, the laser reflected on the shutter reflective blades is sent to another beam block. Beam blocks are not perfect absorber, so we can use advantageously the weak back scattered light from the beam block to calibrate the Pockels cell. In our set-up, a photodiode (PDA36A2, Thorlabs) is thus facing the beam block. Following the shutter, the beam travels though a 2× telescope (GBE02-B, Thorlabs) doubling its diameter. Past the telescope, the beam is then

reflected off several mirrors in a periscope configuration before entering a custom aluminium box housing, mounted to a 95 mm optical rail (XT95 series, Thorlabs), that houses the two scanning mirrors (CRS8K/6215H scanning mirror set, Novanta), shown in blue in Fig 1D. Leaving the scanners, the beam then travels first through the scan lens (green) and tube lens (red), passes through the long pass dichroic mirror of the collection box assembly (yellow) and finally the objective (cyan). The design of these components is covered in the next paragraphs. The steering mirrors in our system are all P01 silver protected mirrors from Thorlabs. Silver mirrors contribute minimally to dispersion which is inevitable when light travels through various optical components. Limiting the dispersion makes it possible to correct for it with the laser's built-in dispersion compensation.

## Optical design: Scan and tube lens

Like for all modern microscopes, infinity-corrected objectives are being used in this design, see Fig 1A. The incident laser light has to be collimated at the back aperture of the objective. For this reason, scan and tube lens must be carefully configured as a telescope. Additionally, to guaranty the laser beam enters the objective back aperture with the same amount of clipping, the scanning mirrors must be positioned in the conjugated plane of objective back aperture. In many commercial infinity corrected microscope designs, the distance between the tube lens and the objective can be chosen within a certain range. In our design, we added the constraint of positioning the objective and tube lens in a telescope, i.e. the distance between them, $d_{TO}$, is the sum of the tube lens and objective's focal length, $d_{TO} = f_O + f_T$. While this has no real impact, it prepares the assembly for possible upgrade where methods such as remote focusing could be used [16–18]. Our design was initiated by choosing a proper focal length for the custom assembly serving as a scan lens. We settled on a focal length $f_S$ = 100 mm scan lens built from two inch diameter components [19]. This choice was influenced by the availability of off-the-shelf lenses and the specific scanning range of the scanners (typically ±10 deg, cf. Fig 1A). To fill the back aperture of the objective with our laser beam, we aimed for a combined magnification of $M = 3.75\times = f_T/f_S$, making the focal length of the tube lens $f_T$ = 375 mm. With these choices, the resulting scanning angles at the back aperture of the objective are ±10 deg/3.75 ≈ ±2.7 deg and with our chosen objective (Nikon 16× with $f_O$ = 12.5 mm) the expected approximate field of view diameter is ≈ 1.2 mm).

The tube lens was also assembled as a lens group from off-the-shelf parts. The relative positions of the components in the two lens groups were optimized in Ansys Zemax OpticStudio. Details are provided below. Fig 2A shows the propagation of light rays for various deflection angles of the scanning mirrors, and how the beams converge on a line under the objective. Note that in our simulation, in the absence of a model for our commercial objective, we replaced it by a perfect geometrical lens. Once the separation between optical elements was optimized, a model was exported in CAD (Autodesk Inventor and Fusion 360), and the optomechanics was designed around it, see Fig 2B. The CAD design and lens groups are highlighted in Fig 2B and 2C (see Fig 1D of a photograph of the same part of the instrument). The key properties that we optimized when designing the excitation system was an essentially flat, and diffraction limited focal point across the deflection angle range of the scanning mirrors. Fig 2D shows an estimate of the two-photon point spread function (PSF-2p), computed as the square of the Huygens PSF in Zemax. Analyzing the PSF-2p across optical deflection angles, Fig 2E, suggests very small field curvature if the objective used behaves close to a perfect geometrical lens. In this simulation, the axial resolution was $\delta z$ = 5.0 μm, and the radial resolution $\delta r$ = 0.75 μm across a wide range of deflection angles.

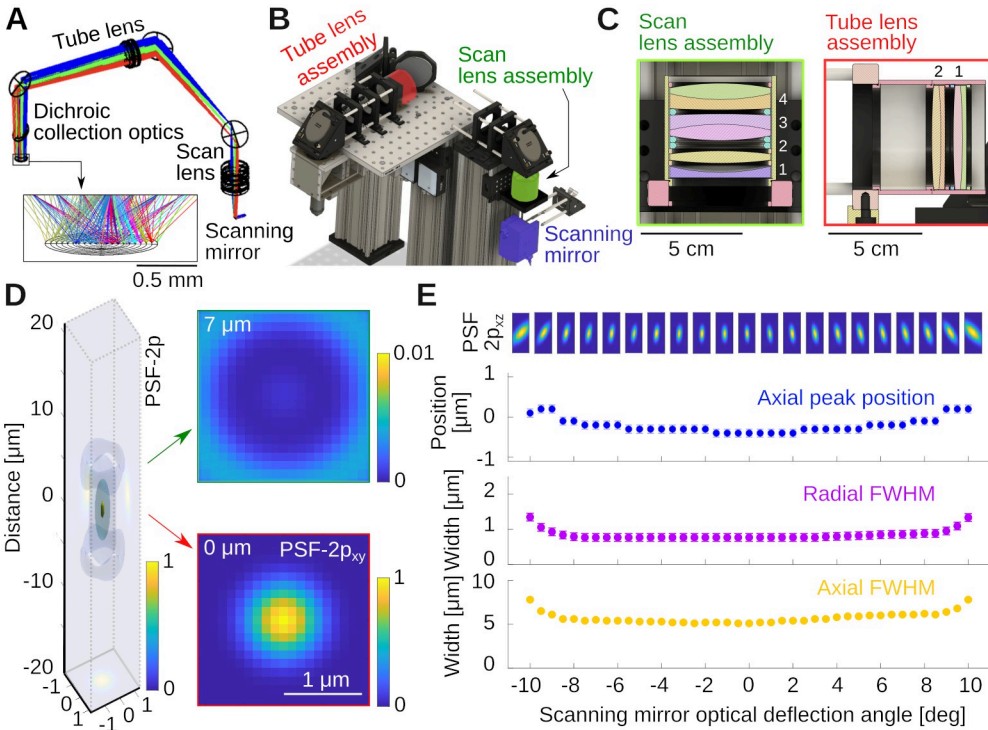

**Fig 2. Optical design of the excitation pathway. A)** Zemax design with the scanning mirror, tube and scan lens groups. Colors indicate light rays produced by different mirror angles of the scanning mirror. Notice how the rays converge in the image plane. **B)** CAD design built around the optical design in A, see also Fig 1C. **C)** Cuts through the scan and tube lens assemblies, which are fabricated from off-the-shelf lenses housed in SM2 lens tubes. The specific lenses are stated in the text. **D)** Simulation of the optical system's two-photon point spread function (PSF-2p) for 0 deg deflection angle, and two example radial sections. **E)** PSF-2p in the xz plane across the full range of deflection angles. Shown below is Petzval field curvature (blue), and radial (purple) and axial (yellow) full-width-at-half-maximum of the PSF-2p. Error bars are resolution limits of the Huygens PSF estimates in sequential mode.

The specific elements that allowed for this performance were as follows: For the scan lens, we used an assembly of four lenses, see Fig 2C, 1: KPC070AR (Newport); 2: LB1199-B (Thorlabs) and 3&4: 2× ACT508–200-B (Thorlabs). The 2× ACT508–200-B lenses are arranged back-to-back and provide the majority of the optical power. The symmetric Plössl design corrects for the odd aberrations: coma, distortion, and lateral colors [20, 21]. The two additional lenses provide corrections to the even aberrations: spherical aberration, astigmatism, and field curvature. We chose the specific lenses and their spacing based on availability from established optics suppliers and the simulation of dozens of combinations in Zemax. The tube lens is inspired by a Petzval lens design [20, 22], consisting of a pair of achromatic doublets with the same orientation, 2× ACT508–750-A, Thorlabs, a design performing better than a Plössl pair as observed by others as well [14, 23, 24].

The two lens groups combined provide a magnification of $M = 3.75×$ to produce a beam that slightly underfills the back aperture of the Nikon NA 0.8 16× LWD objective, effectively reducing its excitation NA to ≈ 0.7. This can be advantageous for *in vivo* Calcium imaging in certain conditions (see Discussion). Of further note, we used a piezoelectrical collar to mount the objective. The collar can move the objective along the optical axis at a fast rate for imaging across multiple planes.

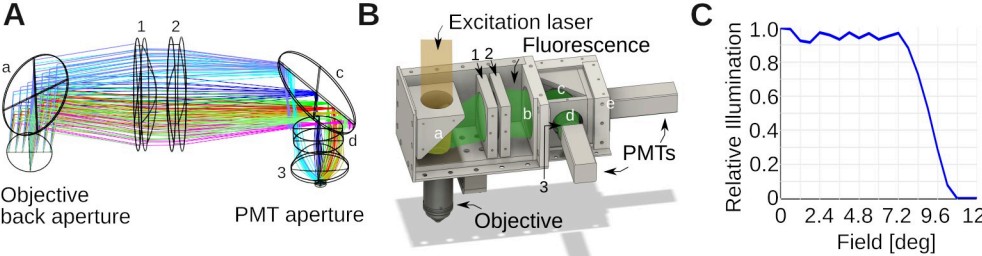

**Fig 3. Optical design of collection system. A)** Zemax design of the collection optics. Colors indicate different emission angles. These rays were computed for the green emission light path. Small letters denote filters: a is a FF665-Di02–40x55; b: a near-infrared blocking filter (FF01–680/SP-50) (not shown in the Zemax simulation); c: FF562-Di03–40x52 and d: FF01–525/45-32 (BrightLine; AVR Optics). Small numbers denote lenses. 1 is a LA1384-A; 2: LB1607-A and 3: ACL25416U-A (all Thorlabs). **B)** CAD design built around the Zemax model in A, with several covers removed to expose the interior. The light paths from the laser and fluorescent emission are indicated in transparent colors. Notice how the dichroic mirror splits fluorescence and excitation light, and directs the former to the photomultiplier tubes (PMTs). The mirrors/filters and lenses are numbered/lettered as in A. Two more filters were added relative to the simulation. b blocks residual IR excitation light λ > 680 nm (FF01–680/SP-50), and e is a FF01–600/52-32 filter for the red channel. **C)** Shown is the angle collection graph of the collection optics, which shows the fraction of collected light as function of the angle it emerges from the objective back aperture. This was simulated in Zemax for the design in A. The collection optics were optimized to collect essentially all light over a ≈ ±8 deg emission angle.

## Optical design: Collection optics

The aim of the collection optics is to bring the fluorescence signal coming out the back aperture of the objective toward the 5 mm diameter photocathode of the Hamamatsu H16201–40 Photomultiplier tube (PMT) module. We incorporated several design principles into our assembly: (1) The collection lens should be large to maximize the collection of as much fluorescent light as possible. This implies that the collection lens should also follow the long-pass dichroic mirror as closely as possible. (2) The collected light should be then guided through an elongated but diameter limited space where another dichroic mirror and filters (Green/Red) will be located. It should therefore propagate as close to parallel as possible from the collection lens axis. This also favors satisfactory performance of the optical filters which are often specified for zero degree incidence. (3) Having the collected photons close to collimated on the aperture of the photomultiplier tubes reduces the sensitivity of the system to surface inhomogeneities of the detector [25–27].

We aimed to record fluorescence from two well-separated spectral ranges, a green (525 ± 25 nm) and a red channel (600 ± 25 nm). Zemax was used to select the lenses of the collection unit and their relative position, while attempting to maximize the collection of light by the detectors. A cartoon is shown in Fig 3A. The colors illustrate different point sources of fluorescent light in the sample plane. Notice the mild divergence, and approximately collimated beams that enter the aperture of the photomultiplier tube. As previously, this design was exported into CAD, and the optomechanics assembled around it, see Fig 3B. Fig 3C shows a simulation in Zemax demonstrating that our design brings nearly all the light escaping the objective back aperture with an angle below ≈8 deg to the aperture of the photomultiplier tubes.

## Mechanical design

Our microscope was mounted on a 16 inch thick optical table. The design was done in CAD Inventor and Fusion 360 (Autodesk), and the results shown in Fig 1C. The aim was a

mechanical assembly that uses as few custom parts as possible. This is relatively easy to achieve for the excitation path, but hardly practical for the collection system which require to be both efficient and light tight. The components shown in the CAD files can be assembled and aligned by an experienced researcher in a few days. When used for teaching, a careful assembly and alignment is viable with about a week of work. When set up on an imperial-unit table, the system can comfortably be built on $4 \times 4 \text{ ft}^2$. The microscope documented here was built on a $4 \times 8 \text{ ft}^2$ table, shared with a second microscope. In metric units, our microscope has a footprint of $\approx 1.5 \text{ m}^2$. In other words, a standard $1.2 \times 2.5 \text{ m}^2$ table can house two of the systems described here. The beam-splitting hardware used for this arrangement is shown in Fig 1B and 1C as well.

The custom mechanical components (e.g. the light-tight aluminium housing of the collection optics) were built in the university's machine shop from aluminium stock or Thorlabs parts. Some parts have tight tolerances which suggests manufacturing in a professional machine shop is preferred. If useful for training, this could however be done by students. An example are the aluminium parts that hold the large dichroic mirrors and optical filters in the collection optics. After drilling the apertures for light to pass through, little metal is left on the part which can potentially lead to warping and distortions if not machined with precision.

Finally, we want to emphasize the difference between a theoretically optimal design in Zemax and finite tolerances in real-world optomechanical parts. It is key for a good microscope design to be robust against such errors. In our design, the objective is fixed, while the mirrors, the tube and scan lens assemblies, and the scanning mirrors can be re-positioned along the optical axis for alignment. This allows the mechanical components sufficient degrees of freedom to optically align all key parts of the microscope. In the supplement, we go through this alignment procedure in detail.

## Scanners power supplies and controllers enclosure

The scanning mirrors are active components. They are controlled by two circuit boards supplied with external power. The driver board of the resonant scanner does not dissipate much heat, but the driver of the slow galvanometer can become warm in normal operation. This heat needs to be dissipated. We mount the circuit boards in a custom electronics box (Gold Box, Acopian), making sure to use heat transfer compound between the circuit boards and the aluminium box, and to mount them securely together with screws. At time of purchase, we requested the controller of the slow galvanometer to be configured for usage at ±28 V (the manufacturer default is ±24 V). In our experience, this improves dynamical properties. The controller requires three supply lines, -28 V, 0 V, and +28 V. To provide these voltages, we used a power supply with two galvanically isolated and adjustable outputs (not every power supply supports this function). The voltage of each of the two channels was set to +28 V. To produce the three supplies, we used these two channels in series. In other words, we connect the positive terminal of the first channel to the negative terminal of the second channel. As the channels are otherwise not connected to a common ground (this is meant with the words "galvanically isolated"), this does not produce a short circuit. Instead, this link effectively provides a new reference point, relative to which the first supply produces a negative -28 V supply, and the second supply produces the +28 V supply. More details are provided in the supplement. The second power supply provides 12 V for the resonant scanner. It is beneficial to choose a power supply that can stably produce these voltages for years. We have made good experiences with Agilent E3630A, and the Instek GPD-3303D. Importantly, double check the power supplies with a good multimeter before connecting the driver boards. Please also see S1 Appendix. This contains illustrated construction instructions and technical details for a detailed illustration.

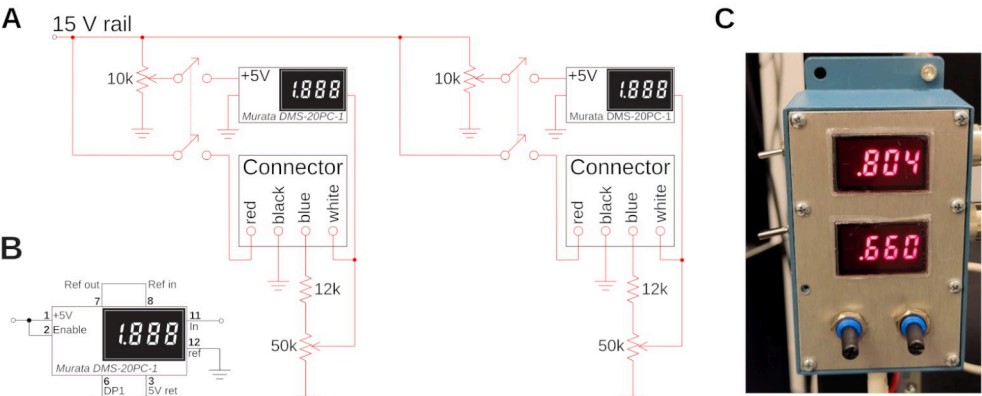

**Fig 4. Custom electronic control circuit and box. A)** Control circuit to set and read the gain of the two H16201P-40 photomultiplier tube modules used for red and green channel. The two modules are connected with two 4-wire connectors, and their current gain is displayed on a small digital panel meter. **B)** Wiring of the panel meter for single-ended configuration. **C)** The circuit in a prototype-box, and mounted to the side of a rack. The two gains for green (top) and red (bottom) channel are visible. A gain of around ≈ 0.8 is typical for Calcium imaging to maximize the faint signal. A gain of ≈ 0.6 is appropriate for very bright samples.

## PMTs control circuit

A simple circuit, see Fig 4A, is used to control and display the photomultiplier gain. The H16201P-40 photomultiplier tube module comes with four cables, two for power supply, and two for gain control. These leads are color-coded. A small circuit for resistance programming of the gain was designed which also displays the gain setting voltage on a small digital panel meter (Murata DMS-20PC). The circuit for controlling two channels (red and green) is shown in Fig 4A. For the tube module, the +15 V supply voltage and ground are provided via the black and red cables. The blue cable is connected to ground via a potentiometer to act as a voltage divider. The gain voltage is then fed back to the module via the white input line, and used for display on the panel meter. A 12k resistor between the PMT module and the potentiometer was added to limit the control voltage which is safer for the PMT. Optimal signal-to-noise ratio for the PMTs is usually ≈0.7 − 0.8 V. The panel meter needs an additional +5 V supply that we obtain from the +15 V via a resistive divider. The meter is operating in single-ended input configuration, see Fig 4B. Due to the simplicity of this circuit, we do not solder this on a circuit board, but rather in a prototype development box directly, see Fig 4C, which is attached to the rig. This circuit needs a well regulated power supply. For convenience, we also use an Agilent E3630A. The PMT outputs were amplified with two Transimpedance Amplifiers (TIA60, Thorlabs), and directly connected to the data acquisition system.

## Ancillary hardware

Our microscope is controlled with ScanImage [28] running on a Windows PC equipped with a data acquisition system (DAQ), both from the same supplier (MBF Bioscience). For the work demonstrated here, we used a fixed-wavelength femtosecond laser systems, based on fiber technology (e.g. [12, 26, 29, 30]). These devices come at ≈1/3 of the cost of a Ti:Sapphire system, and take up significantly less real estate on the optical table. Our laser, an Alcor 920 manufactured by Spark Lasers, was controlled with the same PC as ScanImage through a USB connection. As the beam travels through a number of crystals, lenses, and mirrors with various chromatic properties, significant dispersion is introduced which increases pulse duration,

reducing the two-photon absorption efficiency. It is possible to compensate for the group-delay dispersion of the microscope by giving the short-wavelength components a sufficient head start so that blue and red components arrive at the sample at the same time. We found, empirically, that the built-in group velocity dispersion compensation of the laser was more than sufficient to tune the system to an optimum, which we found close to $\varphi = -20200 \pm 170$ fs$^2$. Details of this measurement, and some theory, are provided below as a teaching example (cf. Fig 9). Regarding light transmission, we found the Spark laser to produce $\approx$ 2.1 W of light. After the half-wave plate and beamsplitter cube, $\approx$ 1.0 W enter the Pockels cell (cf. Fig 1B). The Pockels cell is rotated so that the control voltage produces the largest dynamic range of transmitted laser power (we go through the alignment procedure in depth below). At maximum transmission setting of the Pockels cell, we obtain $\approx$ 260 mW below the objective, a large value, as it is exceeding the maximum recommended power for cortical imaging in mice [31]. All pictures presented in this article were obtained with $\approx$ 15 mW below the objective.

## Ethics statement

Some of the data presented here were obtained from animals. Our work was conducted according to internationally accepted standards and we followed all laws and regulations about animal research in the United States and Germany. We used vertebrate animals in the form of GCaMP6f expressing transgenic Zebrafish (*Danio rerio*) for *in vivo* imaging. Fish were bred and housed at the Princeton Neuroscience Institute. We used five day old larvae immobilized in 2% low melting point Agarose in E3 medium. Larvae were sacrificed after the imaging sessions. As required by law, we have obtained prior approval by the Institutional Animal Care and Use Committee at Princeton University under Protocol Number 2033. Further *in vivo* imaging work was done in the adult fruit fly (*Drosophila melanogaster*). In the United States, work in lower-level invertebrate species like the fruit fly does not require approval by the Institutional Animal Care and Use Committee. We also imaged a previously prepared fixed and stained histological sample obtained from a Wistar rat (*Rattus norvegicus*). This specific sample was made by one of us during their PhD [32]. The animal was bred in the animal house of the Max Planck Institute for Experimental Medicine according to European and German guidelines for experimental animals, and work was carried out with authorization of the responsible federal state authority. Finally, we imaged plant material from the common Dandilion (*Taraxacum officinale*) *in vivo*. This experiment does not require ethics approval in the United States.

# Results

## System performance

Here, we document the performance of the microscope and a few example applications using a Nikon NA 0.8 16× LWD objective (N16XLWD-PF, Thorlabs). We first determined the size and properties of the field of view with a 100 μm grid (R1L3S3P, Thorlabs), imaged using a Fluorescein in water film through a standard #1.5 ($\approx$ 170 μm thick) coverslip, see Fig 5A. The field of view exceeded $\approx$ 1 mm$^2$ ($d \approx$ 1.3 mm along the diagonal), and the field curvature was below the thickness of the thin Fluorescein film $\lesssim$ 10 μm. The maximum scan angle was limited by vignetting of the two inch diameter optics (we discuss this further in the teaching section). When measuring the local scale in μm/px at low magnification across the field of view, we observed small deviations of $\Delta S \approx$ 0.1 μm/px in the center when compared to the edges, while the overall range of magnifications across the field of view was tight, see Fig 5B. The average scale is $S = 2.77 \pm 0.04$ μm/px, which suggests a deviation from uniformity of $\Delta S/S = 0.1/2.8 \approx$ 4%. Next, we imaged a uniform bath of Fluorescein, see Fig 5C. This measures the quality of excitation and the efficiency of the collection optics across the field of view.

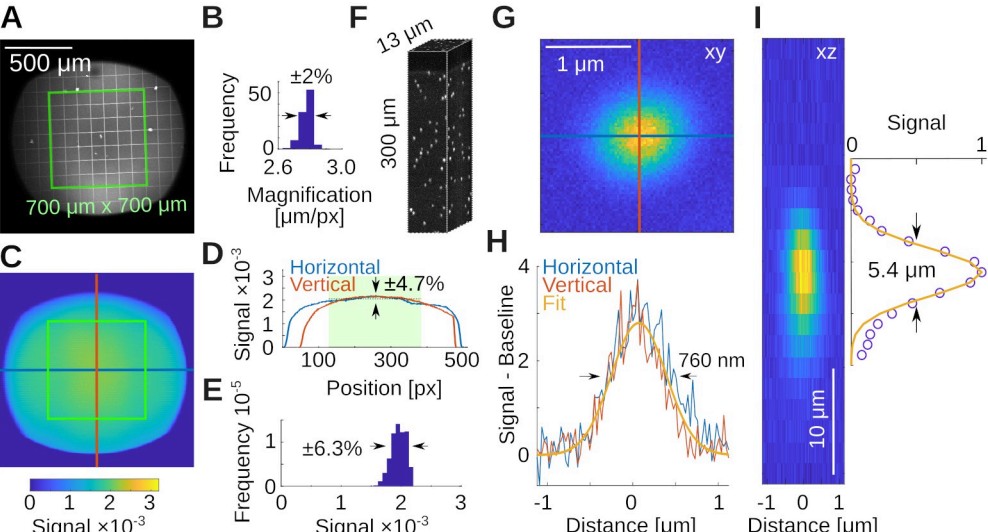

**Fig 5. System capabilities. A)** Measurement of the local magnification with a 100 μm calibration target in a thin Fluorescein film below a coverslip. Shown in green is a $700 \times 700$ μm$^2$ square as reference. **B)** Histogram of the magnifications in A, measured across all squares. **C)** Two-photon intensity measurements from a uniform bath of Fluorescein. Shown are pixel values (maximum is 32768). **D)** Horizontal and vertical profiles through the data in C. The green box is indicated by the shaded green area. Range is ±1 standard deviation. **E)** Histogram of the intensities from within the green box in C. **F)** Volumetric measurement of a 0.2 μm bead sample in 1% Agarose. **G)** Average of $N = 38$ beads. **H)** Horizontal and vertical profiles through data in G and a Gaussian fit reveal a radial resolution of $760 \pm 30$ nm. **I)** Estimate of the axial resolution shows $5.4 \pm 0.9$ μm.

Within the central 700 μm × 700 μm region, the signal deviated from uniformity within ≈ 13% (±1 standard deviation), see Fig 5D and 5E. Following these measurements, we imaged a bead sample of 0.2 μm diameter in 1% Agarose (Dragon Green beads; Bangs Laboratories) to measure axial and radial PSFs. The sample is shown in Fig 5F. Averaging across the $N = 38$ beads in this volume, we measured a radially symmetric full-width-at-half-maximum of $\delta r \approx 760 \pm 30$ nm, see Fig 5G and 5H, and axially $\delta z \approx 5.4 \pm 0.9$ μm, see Fig 5I. The theoretical resolution limits [27] for our system, underfilled to NA = 0.7 at λ = 920 nm and water immersion ($n = 1.33$) were radially FWHM of $\delta r = 0.6\lambda/\text{NA} = 780$ nm and axially $\delta z = 2\lambda n/\text{NA}^2 = 5.0$ μm. This is consistent with our earlier simulations in Zemax (cf. Fig 2), and suggests that our microscope is operating very close to the diffraction limit, and its design specifications.

## Imaging performance

We next imaged a number of representative samples that are readily available to us, namely plant material, a histological and a strongly scattering sample, see Fig 6. The first samples was a Dandilion flower (*Taraxacum officinale*) [33]. Autofluorescence in plant material is a good test to separate compounds fluorescing in the red and green channels respectively, while also exhibiting complex three-dimensional structure [7, 34]. Fig 6A shows a slice through part of the flower, that reveals red-fluorescent structural plant material, together with green-fluorescent Dandilion pollen grains. The complex 3-D structure is readily apparent if one zooms into a small region (white box), and records a stack, see Fig 6B. The pollen grains are hidden under a layer of red-fluorescent plant material as well, which we can easily resolve. Pollen grains are a great sample in a teaching lab, as they are easy to obtain, easy to prepare, and complex when imaged with a two-photon microscope [33]. The next sample is a mounted primary rat cell

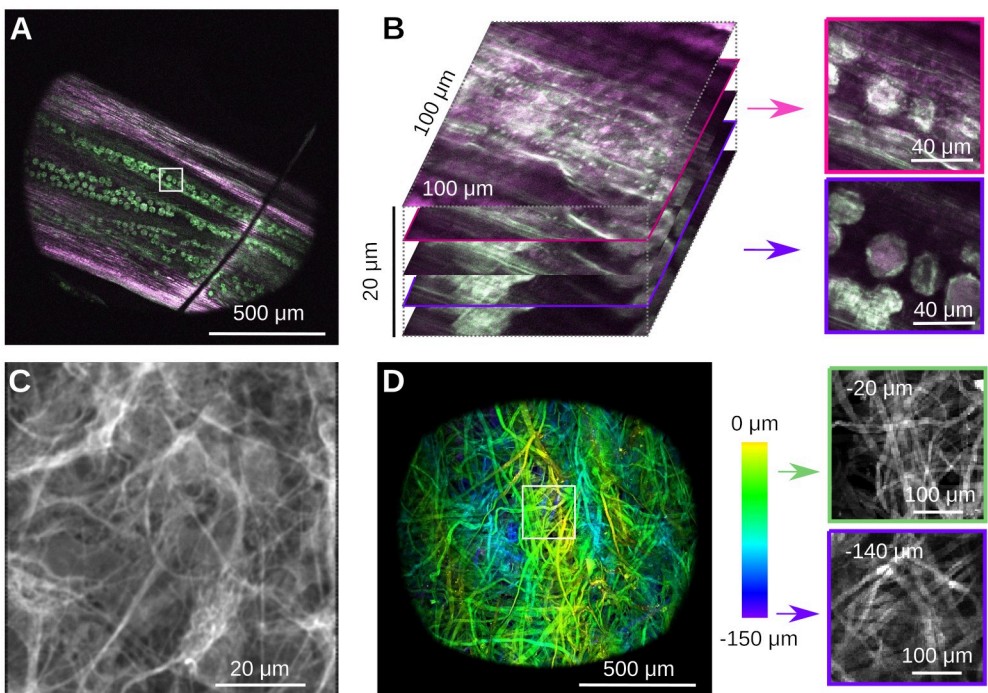

**Fig 6. Example imaging data from simple samples. A)** Autofluorescence of a part of a dandilion flower (*Taraxacum officinale*) *in vivo*. Notice the pollen grains embedded in the plant material. Optical sectioning allows to focus below the plant surface to the grains. **B)** Zoom and volumetric image of a small region (white box) of the sample in A. Optical sectioning shows the surface of the plant and the slice where the pollen grains come into focus. Notice the complex 3D structure of the pollen grains, and their embedding below a layer of red-fluorescent plant material. **C)** Histological sample showing a stain for glial fibrillary acidic protein in rat astrocytes. **D)** Image of a cotton napkin (Kimwipe) with depth coded as color. Zooming into a small region (white box) shows how strands of cotton are woven together, and how the pattern changes with depth. Notice how image quality deteriorates in this strongly scattering sample.

culture [32], stained for GFAP (Rabbit-Anti-GFAP, ab33922) in the green channel (Donkey-Anti-Rabbit Alexa Fluor 488, ab150061), see Fig 6C. This demonstrates the complex structure of glial fibrillary acidic protein in astrocytes. Finally, we imaged a piece of a cellulose napkin (Kimwipe, Fisher Scientific), see Fig 6D. This is an example of a strongly scattering sample with complex 3D fibrous structure. To demonstrate this structure, we acquired a volume and color-coded depth. Individual slices at different depths reveal uncorrelated fiber structures. In contrast to our earlier experiments in a less scattering sample (cf. Fig 5), the image quality here deteriorates with depth.

Next, we imaged neural activity *in vivo* in both larval zebrafish (*Danio rerio*) and adult *Drosophila melanogaster*. For zebrafish imaging, we used the HuC:H2B-GCaMP6f line of transgenic Zebrafish [35]. Neurons in this organism express the fluorescent protein GCaMP6f in their nuclei whose green fluorescence strongly depends on the cellular concentration of Calcium ions. We imaged a five day old larva immobilized in 2% low melting point Agarose in E3 medium, see Fig 7. We first performed a volumetric scan of the organism, Fig 7A shows a rendering in Fiji [36], and then focused on a plane with a large number of visible neurons, Fig 7B. In a small region, we used our piezoelectric collar to collect volumetric data from three planes, see Fig 7C. We processed these data with Suite2p [37], which identified many neurons with complex calcium transients, Fig 7D.

To image in the adult fruit fly, we head-fixed the fly in a custom 3D-printed mount [38] (Fig 8A). We then perform a dissection which removes the cuticle from the back of the head of

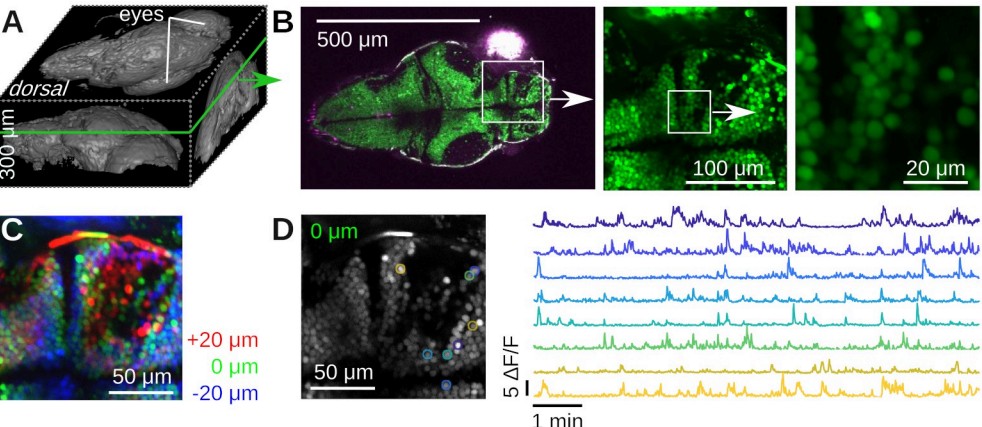

**Fig 7. Example calcium imaging in transgenic Zebrafish (*Danio rerio*). A)** 3D rendering of a 300 μm deep volumetric image of a five day old Zebrafish larva with Fiji [36]. **B)** Dorsal slice through the same animal and two zoom views down to cellular scale. **C)** Color-coded ≈ 10 min of time-averaged volumetric imaging data of three 20 μm-spaced planes using the piezoelectric collar attached to the objective. The layers are colored in red/green/blue. No motion correction was applied. Notice the blurry averages, caused by motion of the fish. **D)** Running Suite2p on the data in C produces numerous regions of interest (ROIs) with complex Calcium dynamics. Left: Suite2p applies motion correction, resulting in a much crisper average picture. Right: Time series of a few ROIs over 10 min.

the fly, providing optical access to the brain. We imaged 6-day-old transgenic adult flies which express GCaMP6s and myr-tdTomato pan-neuronally, labeling all neurons cytoplasmically. We volumetrically imaged a volume approximately $300 \times 350 \times 50$ μm$^3$ in scale, spanning a fraction of the central brain of the fly (Fig 8B). We then motion-corrected and segmented the GCaMP6s volume into regions of interest (ROIs) with spatial clustering algorithms [38], revealing diverse spontaneous calcium activity (Fig 8C).

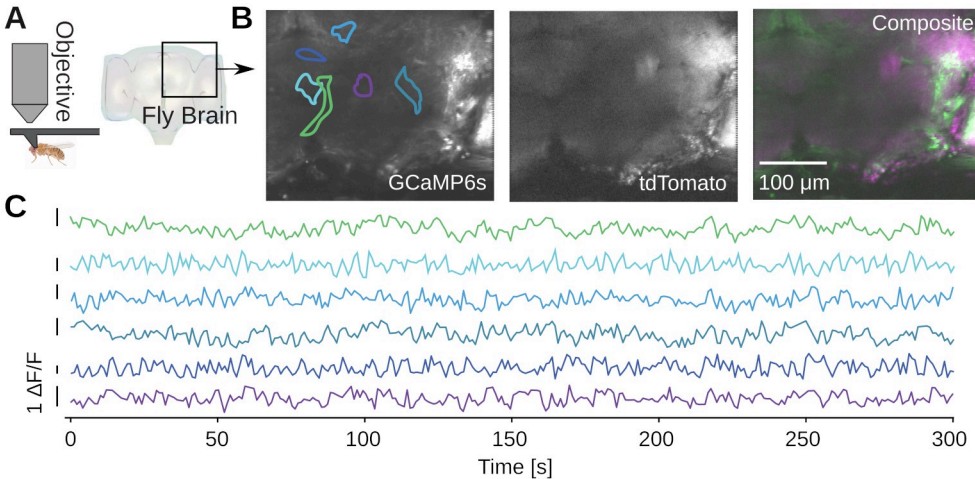

**Fig 8. Example calcium imaging in adult *Drosophila melanogaster*. A)** Imaging from the brain of a head-fixed adult fly [38]. A dissection is performed to remove the cuticle from the back of the fly's head and provide optical access to the brain, and the imaged brain volume is outlined. The transgenic flies express pan-neuronal GCaMP6s and myr-tdTomato, driven by the nsyb promoter. **B)** Average projections over ten 5 μm-spaced planes of both imaging channels, acquired at a temporal resolution of 1 volume/s. XY boundaries of exemplar ROIs are indicated. **C)** The channels are motion corrected and spatially clustered ROIs are extracted. Examples of spontaneous calcium activity extracted from ROIs in the brain volume.

## Teaching

Twinkle was assembled, disassembled, and reassembled three times by different groups of researchers and students. In the following paragraphs, we summarize our experiences, and produce two examples for what we consider useful learning experiences for two-photon microscopy. The detailed building instructions and additional information are available the supplement, S1 Appendix. This contains illustrated construction instructions and technical details.

To organize our workshop, we freed around 2 weeks in the summer. On the first day, we provided an introductory lecture going through the principles of two-photon microscopy (cf. Fig 1A), safety, and the plan ahead. Next, we completely stripped the optical table and returned Twinkle's components back into their containers. Then reassembly began, first setting up and aligning laser and beam-splitting hardware, then the Pockels cell, the shutter, telescope, and associated beam blocks (cf. Fig 1B and 1C). This was done over the course of days 1–3 during which students became familiar working with safety goggles, and aligning lasers using a power meter and cards. In particular, the alignment of the Pockels cell requires care, as its orientation has to be carefully set to produce the largest dynamic range of transmitted laser power depending on the control voltage (see teaching examples below). Following the conditioning of the beam, we then proceeded to assemble the mechanics of the microscope head (cf. Fig 1D). Reassembling the box housing the scanning mirrors, the lens groups from individual elements, and the collection optics was done from days 3–6 (cf. Figs 2–4) After this was done, we assessed the performance of the instrument and fine tuned it on days 7–8 (cf. Fig 5). When properly aligned, we shielded the optical path with sheet metal and Thorlabs tubing, and imaged various samples on day 9 (cf. Figs 6–8). On the last day, day 10, we addressed questions that came up during the course of the workshop.

To exemplify a simple teaching example, see Fig 9. Fig 9A shows light transmission through the Pockels cell for two different orientations (red and blue) to teach the working principles of the Pockels effect. The optimal angle is shown by the blue dots. A fit to the expected transmitted power curve,

$$P(V) = P_0 \sin\left(2\pi \frac{V}{V_0} + \phi\right)^2 \tag{1}$$

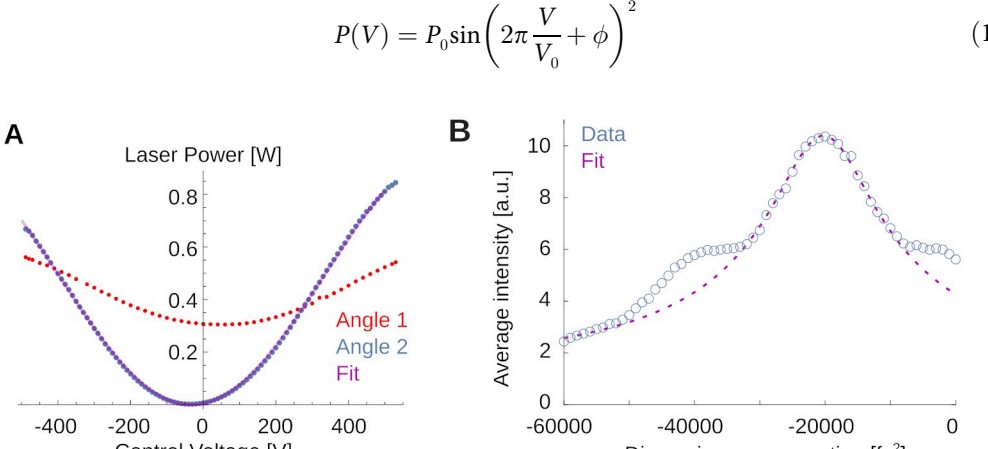

**Fig 9. Examples for teaching principles of two-photon microscopy. A)** Transmitted laser power as function of the control voltage applied to the Pockels crystal for two different crystal orientations. The red orientation is far from the optimum and produces a small dynamic range for intensity control. The blue orientation is close to the optimal. The thin purple line is a best fit (see text). **B)** Fluorescence in Fluorescein as a function of applied dispersion compensation. A fit allows to estimate the pulse width. Note also the symmetric shoulders not captured by the fit, suggesting a non-Gaussian pulse shape.

is shown as thin purple line. Deriving this from geometric principles, and aligning the Pockels cell on the table for greatest dynamic range is relatively straightforward and the students perform these measurement by scanning through voltage settings on the Pockels driver, while monitoring light intensity. This explains to them the optimal orientation of the crystal relative to the laser.

If students are interested in dispersion of the laser pulse and its correction, a more challenging experiment and analysis are shown in Fig 9B. Here, students used the fluorescence at fixed laser power for different values of the dispersion compensation to infer the laser pulse width without the use of an additional autocorrelator. This is possible because the efficiency of two-photon fluorescence depends inversely on the pulse duration [1, 3]. The laser's built-in dispersion compensation can give the short-wavelength components a head start such that after delays in the optical path, blue and red components can arrive at the sample at the same time. One can write down the expected fluorescence $I$ as function of this group delay dispersion $\varphi$ applied to a $\varphi_0$-dispersed Gaussian pulse of FWHM width $\Delta\tau$ as

$$I(\varphi) = I_0\left(1 + \alpha\frac{(\varphi + \varphi_0)^2}{\Delta\tau^4}\right)^{-1/2}. \tag{2}$$

Here, $\alpha = 16\log(2)^2$. This equation can be derived by expressing the Gaussian pulse shape in the Fourier domain, applying a phase-shift along the laser path to quadratic order in the frequency, and computing the result in the time domain. Deriving this expression is a fun at-home exercise. The fit suggests that the various optical components introduced a dispersion of $\varphi_0 \approx 20200 \pm 170$ fs$^2$. When removed with the built-in dispersion compensation of the laser, our system operates with a FWHM pulse of $\Delta\tau \approx 150 \pm 20$ fs, which is close to, but significantly longer than the $\approx 110$ fs pulses produced by the laser itself (measured with an autocorrelator). This suggests detectable non-linear dispersion that we cannot compensate with our optics. Note that analyses like these, relating laser physics and fluorescence, are close to the research frontier [39, 40].

## Discussion

In this article, we have covered the design, building, and alignment of a two-photon microscope. We then assessed its performance and demonstrated several possible use cases, from plant physiology over material science to neuroscience. In the following paragraphs, we will expand on some recent scientific use cases of two-photon microscopy, discuss improvements to the design, elaborate on its use for education, and conclude with a note about our instrument in the growing space of open source scientific instrumentation.

### Scientific use cases

Based on its unique ability to image with high resolution and limited photo damage deep into scattering samples, two-photon microscopy became an important tool across the life sciences. For example, *ex vivo*, two-photon microscopy can monitor, in three dimensions, both the living and cornified keratinocytes of the epidermis, the collagen/elastin fibers in the dermal layer of the skin [41] or the dynamics of quiescent skeletal muscle stem cells embedded in the muscle at cellular resolution [42]. *In vivo*, two-photon microscopy allows to monitor the dynamics of viral infections in the lung [4], cell vitality and apoptosis, fluid transport, receptor-mediated endocytosis, blood flow, and leukocyte trafficking in the kidney [5], blood vascular hemodynamics in the thymus [43], stem cells in the skin [44], produces new insights into cancer *ex vivo* and *in vivo* [6], and even entered clinical practice [45].

Arguably, two-photon microscope is particularly transformative in systems neuroscience because it facilitates the non-invasive measurement of fast neuronal dynamics with single cell resolution in awake and behaving animals [10, 46]. In the mouse, *Mus musculus*, two-photon microscopes have been used to classify the function and neural inventory of the retina [47], study sequential activity in cortex [48] and subcortical structures [49], led to transformative insights into signals in the dopaminergic reward system [50], and revealed an organized map among grid cells [51]. In non-human primates, two-photon microscopes have been used as optical brain computer interfaces [52], and to survey the spatial organization of motor cortex in reach movements [53]. They also have potential to further our understanding of the function of the primate retina [54, 55]. In the Zebrafish larva, *Danio rerio*, two-photon microscopy can image the entire brain with single cell resolution [56]. Two-photon microscopy has also become a standard method for monitoring neural activity in invertebrates. In the fruit fly, *Drosophila melanogaster*, imaging from sparse, genetically specified neuron populations [57, 58] has enabled many circuits to be functionally mapped, including auditory [59], courtship [60, 61], and navigation circuits [62]. Two-photon microscopes have also enabled volumetric pan-neuronal imaging, in which large portions of the whole fly brain can be recorded from with high temporal resolution [38, 63]. In *Caenorhabditis elegans*, it allowed to measure an atlas of neural signal propagation [64].

New inventions are published regularly, such as the incorporation of adaptive optics to improve the signal amplitude [19], the combination of two-photon imaging with two-photon stimulation for all-optical interrogation of neural circuits [65–67], and extremely large fields of view for mesoscopic imaging across brain areas [18]. Recently, several approaches have also been developed to increase the size of the volumes being imaged beyond the limits set by the traditional sampling strategy while keeping the time resolution constant. New approaches probing extended regions rely on signal demixing to reconstitute the underlying structure [31, 68]. Another approach uses a set of axially separated and temporally distinct foci to record the entire axial imaging range near-simultaneously [69].

## Improvements/Modifications to the microscope

(1) We used silver mirrors to direct the beam. These reflect around 98% of incoming light. Dielectric mirrors perform significantly better, typically exceeding 99.5% reflectivity. With around 8 to 10 mirrors in the beam, this can boost overall transmission by 10%-20%, but comes at slightly higher cost and can introduce complex dispersion [70]. In our experiments, we have not encountered power limitations: For the data presented here, we used only $\approx 15$ mW under the objective while at maximum transmission the system allows for up to $\approx 260$ mW to leave the objective. For simplicity, and similar to other designs [18], we therefore used exclusively silver mirrors. However, for stimulation or ablation experiments, considerations might be different.

(2) In our design, a piezoelectric collar is attached to the objective. Physically moving the objective along the optical axis allows to image several planes for fast volumetric imaging *in vivo* or tissue samples *ex vivo*. The time required by the piezoelectric collar to stabilize can reach the scale of 10 ms when using large and heavy objectives. In these cases, volumetric imaging is best achieved using a ramp where the objective is constantly moving at the same speed, rather than doing steps of constant height for every planes. An electrically tunable lens (ETL) [71] or remote focusing [16] are an alternative to the piezoelectric collar. These techniques are a good alternative when the objective is difficult to move (such as a very heavy objective) or cannot be moved, for example during volumetric imaging combined

with controlled stimulation, as moving the objective would also change the location of the stimulation spots.

(3) Estimating the pulse duration suggests non-linear dispersion that we cannot correct. This is highly dependent on the laser. For readers interested in this aspect of microscope design, we refer to [12, 40]. If available, other light sources might be worth exploring.

(4) We made the design choice to use a resonant and slow scanning mirror pair positioned close together in the same mount. Rather than the mirrors themselves, it is the plane equidistant between the rotational axis of both scan mirrors that is conjugated to the back aperture of the objective. This is a compromise that is used by many (e.g. Thorlabs' Cerna or earlier designs [13, 34, 72]). Alternative microscope designs add a pair of lenses between the slow and resonant scanning mirrors to position them in conjugated planes. Our design choice significantly simplifies the design and reduces the amount of glass between laser and sample to reduce dispersion. It does, however, introduce a slight Barrel distortion that becomes visible at low magnifications.

(5) The fixed 2× beam expander can be replaced with variable/zoom beam expander that allows a user to dial in a particular beam magnification to tweak the point spread function for a particular application [73]. While a high numerical aperture is beneficial for high axial resolution and improved separation of somatic and neuropil signals [74, 75], the large angles at which excitation light is brought into the brain results in longer paths for the light to take. This can lead to more scattering. For experiments aiming to image deep in the brain, a lower NA can be beneficial [76]. Similarly, the presence of axial brain motion of several μm in awake and behaving animals can introduce movement artifacts at high numerical apertures [77]. That said, even when underfilled by the excitation beam, a large NA of the objective is beneficial to collect as much fluorescent light as possible.

(6) When imaging fixed samples, movement artifacts are not an issue. For such experiments, particularly *ex vivo* work, it would be strategic to fully fill the back aperture of the objective and produce a PSF that is as tight as possible.

(7) Using an objective with an even larger numerical aperture can significantly increase collection efficiency. Our objective choice was made with long working distance in mind. The 16× water-immersion objective with a numerical aperture of 0.8 accepts around 10% of the $4\pi$ square radiants of emission. A possible alternative, the Nikon 25× with a slightly smaller 2 mm working distance, offers a numerical aperture of 1.10 and would collect 22% of light, an improvement of 2.2× to collection efficiency.

(8) Our microscope exhibits (small) field curvature. In neuroscience, field curvature is often not considered critical because samples are rarely flat. If the microscope was used for precision length measurements, for example in *ex vivo* tissue samples or for lithography purposes, one would want to optimize field flatness. Very small field curvature can be obtained with commercial f-theta scan lenses.

(9) Resonant scanning mirrors are fast, but cannot be steered. If a user was interested in imaging in custom regions of interest, the scanning mirror pair could be replaced with a pair of slow galvanometic mirrors, which can be controlled with high precision. Alternatively, a third scanning mirror can be used to move the imaging area to a different location. The third mirror would need to ne conjugated to the resonant scanner. These scan heads are currently available commercially, e.g. the Vidrio Rapid Multi Region Scanhead.

(10) For some applications, it can be beneficial to combine the two-photon system with wide-field imaging. To this end, the two inch mirror above the collection optics could be replaced with a 800 or 850 nm short-pass dichroic mirror followed by a tube lens and a camera. This would allow infrared (IR) imaging around 700 nm through the collection optics, as it is transparent above 665 nm. Infrared illumination can be provided with an LED. While this is relatively straightforward to implement [13], our microscope design is not compatible with the implementation of wide-field fluorescence.

(11) Our fluorescence filters have an average optical density $> 4$ in the range between 400 nm and 460 nm. This is advantageous for use in virtual reality systems, because light leaving a projector can be filtered to this range before projected on a dome surrounding the objective and animal. We have successfully used budget friendly colored glass filters in front of the projector (Schott BG25 or Schott BG5) to block projector light from entering the excitation path [77]. Combined with a light shield around the objective [78], such a projection system will only minimally interfere with imaging (e.g. [49]). This is a useful range for visual experiments in mice, but if other model organisms are desired, the optical filters need to be adjusted accordingly.

(12) Parts of our microscope can be replaced with commercial products. For example, we have used commercial telecentric scan lenses in the past [31, 79] in combination with a three inch diameter tube lens. These come at higher cost and eventually introduce higher dispersion, but exhibit very low field curvature on a very large field of view. While low field curvature is not always considered the highest priority in neuroscience applications, this can be a useful substitution.

(13) To aid mechanical stability, the aluminium plate forming the base of the head of the microscope could be replaced with a steel plate, or a thicker piece of aluminium, that is less prone to vibrations when bumped into. This would also be important if one wanted to extend the microscope head to provide even more space around the objective.

## Teaching advanced methods in neuroscience

In addition to training researchers on how to set up a two-photon microscope, we aimed to provide examples of imaging data that are not neuroscience related, and do not require the handling of animals or living neurons (cf. Figs 5, 6 and 9). Other samples are possible as well (such as cheese [34]). Working with such samples can make experiments in a teaching lab easier as such spaces are not always designated animal work areas. If students can work with animals, the instrument is ideally suited for various neuroscience experiments. For example, the Calcium imaging demonstrated in Figs 7 and 8 could be combined with pharmacology to demonstrate changes of neural activity patterns. Or students can learn in a controlled environment how to get optical access to the nervous system and hone in their surgery skills to produce good quality optical windows [46, 80]. Our microscope was set up in a teaching lab space at the Princeton Neuroscience Institute. It was disassembled and re-assembled by various generations of students and postdocs. We found that assembling and aligning the microscope, as described, takes around 10 work-days. These two weeks seem to be a good compromise between depth, and researcher commitment. Various additional content across difficulty levels can be added depending on student interest and abilities (cf. Fig 9). For example, and only mentioned in passing before, the history and current applications of the two-photon effect and its use in microscopy is a fascinating subject to explore, e.g. [19, 81].

Students might also become interested in further optimizing the optical design, or wonder why many lenses are doublets and used in lens groups. Some of these questions have intuitive answers. Doublets, for example, contain more glass surfaces that can be optimized to produce a better performing lens when compared to the two surfaces of a singlet [82]. To our knowledge, the first mentioning of the minimization of aberration by placement of several subsequent doublets was done by Joseph Lister in 1830 [83]. This idea was used for many designs, including pairs of doublets in Plössl and Petzval eye pieces, or objectives themselves [84]. Other questions can sometimes be linked to prior student knowledge. For example, for students interested in astronomy, the tube lens can be thought of as an eye piece with a long focal length and eye relief [22]. In our system, the tube lens projects the image formed by the scan lens to an exit pupil, the back aperture of the objective. The objective then focuses the collimated beams onto the sample. In astronomy, the eye piece forms collimated beams of the image formed by the objective lens. The beams converge at some distance (the "eye relief") to enter the pupil of the eye of a human observer. The lens in the human eye then focuses the collimated beams onto the retina. The aforementioned Plössl designs are known to perform particularly well for long eye relief in astronomy [22, 85], and are a common design in two-photon microscopes as well (e.g. [86]). Our design is close to this, but flipped one of the two lenses to a Petzval configuration. In our hands, this configuration performed better [14, 23, 24].

For students interested in photography, the scan lens can be thought of as a 100 mm f/2 camera lens. Camera lenses face similar challenges as our scan lens. They have to focus distant objects, at different angles, into the same flat photographic emulsion, while faithfully depicting true angular distances with minimal aberrations. Think for example about stars on the sky, whose light arrives as beams at different angles at the camera lens. Addressing this problem lead to lens designs that were very similar to our scan lens. One important design, sometimes referred to as a Gauss doublet objective [20], features two symmetrically configured doublets providing much of the optical power, while additional lenses control aberrations. For example, the Zeiss Planar lens from 1897 features 6 elements in 4 groups with a symmetric pair of doublets in a design very similar to our scan lens assembly [87]. To help students and researchers explore these sometime challenging subjects theoretically, we provide all Zemax files for the microscope optics. More advanced optimization in Zemax requires the use of their programming API. For example, the simulated point spread functions and their analysis in Fig 2 were computed in Matlab, using the Zemax API. The code is provided in our repository [11]. This allows for fast optimization of system parameters by effectively scanning through lens spacing, lens types, and the like. This can allow researchers and students to explore the limits and possible future improvements of the system. Experimental modifications of the system are simpler in the format of a workshop. For example, students can swap or rotate lenses and observe the effects. The scan lens assembly is particularly sensitive to this, because the incoming laser beam is concentrated into a small region that gets scanned across the surface of the lens. Small surface imperfections will introduce noticeable differences across spot positions, and significantly affect field flatness, cf. Fig 5C–5E. Small misalignment in the optical path can also lead to characteristic image artifacts, for illustrations see [88, 89], that can help to develop an intuition about how the system operates.

Finally, it can be a useful learning experience to re-use parts from other microscopes, such as XY scanners from a confocal microscope [90], and to mix and match parts to learn what works and what does not. It is an important learning experience to understand failure modes. For example, assembling the lens groups can lead to dust when tightening the retaining rings through the anodized aluminium threads. Or the rectangular "TV screen" field of view (visible in the figures at zoom level 1 above) is caused by a round, and not elliptical mirror above the scan engine. This mirror clips the beam along one axis. (This is corrected in

the CAD files.) When analyzing data, we suggest to use simple programs. Our image analyses here were done in Fiji [36] combined with simple python and matlab scripts. In our experience, students learn very effectively through hands-on open-ended pedagogy and pursuing their own interests [91].

### Open scientific instrumentation

In this article, we summarized the instrument's mode of operation, and provided a design overview. For complete and illustrated building instructions, see the attached S1 Appendix. This contains illustrated construction instructions and technical details and our Online Repository. This contains all the underlying materials of this article. To aid discoverability, we have generated an Open Know How Manifest (OKH-manifest), located in the Online Repository. Our project is also certified by the open source hardware association (OSHWA UID US002677).

The growing space of open instrumentation can improve science in several ways: (1) enabling faster innovation through lower costs and enabling work in low-resource settings, (2) facilitating review and inspection by avoiding black-box designs and (3) providing a benchmark for innovation towards next-generation technology [89]. We hope that our instrument, combined with earlier designs [13, 14, 72], and recent developments, like the head mounted microscope [92], or the Janelia Research Campus microscope [15], can serve as a benchmark for future instruments. Relatedly, we have contributed to other projects that follow this philosophy in microscopy [93], but also in mechanical ventilation during the COVID-19 pandemic [94, 95], and a μ-contact printing device [96]. These experiences have informed this article's narrative, the design and organization of the published files, and also the choice of an open access journal. We hope that our open approach can make two-photon microscopy more accessible to researchers, thereby greatly expanding the ability to train students and producing new inventions. Our ventilator work has led to one of the first devices using deep reinforcement learning that performed superior to classical control schemes [97]. Future will tell how our community will make best use of Twinkle. The world is moving towards more open technology [98], in particular in the microscopy community [99]. We hope that our microscope and its future iterations can play a part in this movement.

### Supporting information

**S1 Appendix. This contains illustrated construction instructions and technical details.** A `.pdf` file with numerous pictures, annotations, and detailed step-by-step building instructions for the microscope. The raw code, pictures, and vectorized versions of the figures are available in the online repository. **Online Repository. This contains all the underlying materials of this article**. A repository, hosted on github, including code, `.svg` figures, and the raw `.tex` files for this article and its supplement. In addition, we provide all CAD and optical design files, material billing and other useful information. The CAD files are provided in several standard formats. The optical design was performed in Ansys' OpticStudio (Zemax): https://github.com/BrainCOGS/Microscope. A static DOI for the materials used in thie article is available on Zenodo [11].
(PDF)

### Acknowledgments

We thank Beatrice Hadiwidjaja for the transgenic zebrafish husbandry and Drs. Lindsay Collins and Anthony Ambrosini for hospitality and accommodating our tinkering in the teaching

lab of the Princeton Neuroscience Institute. We thank Kai Bröking and Peter Rupprecht for helpful comments on the manuscript.

**Disclaimer**: The aim of our work is to allow anyone to build a high-performance and cost-effective two-photon microscope. However, the information here cannot be exhaustive. We recommend anyone embarking on this journey to engage with the community, to carefully think about effective solutions to their use-case and specific experiment at hand, and to adjust to the ever changing landscape of available technology. When adapting the optical path to their needs, we suggest starting at our lens combinations and then optimizing (at the very least) the lens spacings in Zemax. Technology develops rapidly and many of the specific components mentioned here will be outdated in a few years. These parts should be updated as needed. We hope that the narrative in this article helps to make such future replacements straight forward.

## Author Contributions

**Conceptualization:** Stephan Y. Thiberge.

**Data curation:** Manuel Schottdorf, E. Mika Diamanti, Albert Lin, Stephan Y. Thiberge.

**Formal analysis:** Manuel Schottdorf, E. Mika Diamanti, Albert Lin.

**Funding acquisition:** Stephan Y. Thiberge.

**Investigation:** Manuel Schottdorf, P. Dylan Rich, E. Mika Diamanti, Albert Lin, Sina Tafazoli, Edward H. Nieh.

**Methodology:** Manuel Schottdorf, P. Dylan Rich, E. Mika Diamanti, Albert Lin, Sina Tafazoli, Edward H. Nieh, Stephan Y. Thiberge.

**Project administration:** Manuel Schottdorf, Stephan Y. Thiberge.

**Resources:** Manuel Schottdorf, Stephan Y. Thiberge.

**Software:** Manuel Schottdorf, E. Mika Diamanti, Albert Lin.

**Supervision:** Stephan Y. Thiberge.

**Validation:** Manuel Schottdorf, P. Dylan Rich, E. Mika Diamanti.

**Visualization:** Manuel Schottdorf, E. Mika Diamanti, Albert Lin.

**Writing – original draft:** Manuel Schottdorf.

**Writing – review & editing:** Manuel Schottdorf, P. Dylan Rich, E. Mika Diamanti, Albert Lin, Sina Tafazoli, Stephan Y. Thiberge.

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
