## [Decision Letter · Decision Letter 0]

10 Nov 2024

PONE-D-24-42429TWINKLE: An open-source two-photon microscope for teaching and researchPLOS ONE

Dear Dr. Schottdorf,

Thank you for submitting your manuscript to PLOS ONE. After careful consideration, we feel that it has merit but does not fully meet PLOS ONE’s publication criteria as it currently stands. Therefore, we invite you to submit a revised version of the manuscript that addresses the points raised during the review process.

The reviews have expressed a positive evaluation of the manuscript, and it has been recommended for minor revisions. You can find more detailed feedback and suggestions from the reviewers below to guide any necessary changes. 

We look forward to receiving your revised manuscript.

Kind regards,

Luca Pesce, Ph.D.

Academic Editor

PLOS ONE

Journal Requirements:

“MS and MD are supported by NIH grant U19NS132720. MS is also supported by a C.V. Starr fellowship and a Burroughs Wellcome Fund’s Career Award at the Scientific Interface. AL was supported by the NSF through the Center for the Physics of Biological Function (PHY-1734030). We thank Beatrice Hadiwidjaja for transgenic Zebrafish husbandry, and Drs. Lindsay Collins and Anthony Ambrosini for hospitality and accommodating our tinkering in the teaching lab of the Princeton Neuroscience Institute. We thank Kai Br¨oking for helpful comments.”

“MS and MD are supported by NIH grant U19NS132720 (https://www.nih.gov/). MS is also supported by a C.V. Starr fellowship and a Burroughs Wellcome Fund's Career Award at the Scientific Interface (https://www.bwfund.org/). AL was supported by the NSF (https://www.nsf.gov/) through the Center for the Physics of Biological Function (PHY-1734030). The funders did not play any role in the study design, data collection, analysis, decision to publish, or preparation of the manuscript.”

Reviewers' comments:

Reviewer's Responses to Questions

**Comments to the Author**

1. Is the manuscript technically sound, and do the data support the conclusions?

Reviewer #1: Yes

Reviewer #2: Yes

2. Has the statistical analysis been performed appropriately and rigorously? 

Reviewer #1: N/A

Reviewer #2: N/A

3. Have the authors made all data underlying the findings in their manuscript fully available?

Reviewer #1: Yes

Reviewer #2: Yes

4. Is the manuscript presented in an intelligible fashion and written in standard English?

Reviewer #1: Yes

Reviewer #2: Yes

5. Review Comments to the Author

Reviewer #1: The manuscript reports “Twinkle”: a microscope for Two-photon Imaging in Neuroscience, and Kit for Learning and Education, which is described as a fully open, high-performance and cost-effective research and teaching microscope. The text is well written and easy to follow also for more inexperienced scientists, and the basic idea can be of interest. The supplementary guide is very detailed and I find it very useful. However, I find some criticalities.

The major one is that the overall cost for all the instrumentation is not much cheaper than many other custom-made two-photon microscopes available in optic laboratories, especially since the price does not consider the costs for laser, the most expensive component. Therefore, I do not see a novelty in this, that however I find more on the ease of production and the educational nature of the paper.

My first suggestion is to introduce more details on how the design of the setup can be improved/modified based on other specific uses that one can need it for, as the main purpose of the paper is to be educational.

The following are questions that arose while reading the manuscript and may suggest additional information or clarifications that could strengthen the paper.

1. Please mention, in the introduction section, how this setup can be applied not only in neurosciences, but also in other fields of study.

2. I suggest to provide an estimation of the budget required to purchase a Ti:S laser, as this can dramatically change the final costs.

3. Line 82: please provide an estimation of the distortion percentage.

4. Line 272: please provide an estimation of the cost.

5. Line 324: please reformulate the sentence “Confident that the system operates as intended”.

6. In the discussion session, please provide an overview of the possibility to use the setup to perform ex-vivo studies on explanted organs.

Reviewer #2: In this manuscript, the Authors describe a two-photon laser scanning fluorescence microscope that is cost-effective, easy to build, and well-suited for both didactic and scientific imaging purposes. Additionally, the Authors demonstrate how this open-source microscope can be utilized as a teaching tool.

The manuscript is well-written and scientifically sound, making it a valuable contribution to the bioimaging community. I particularly appreciate how the manuscript covers technical aspects in a manner accessible to students and early-career researchers while maintaining a level of rigor that more experienced researchers will also find inspiring and thought-provoking.

I recommend the Editor accept this manuscript for publication, pending very minor revisions.

Specific suggestions for revision:

1. The in-series configuration of the two-channel power supply to provide +28 V and -28 V should be described in greater detail. For instance, the connections needed to generate these opposing voltages, along with the shared ground, should be clearly explained. While this information is briefly mentioned in the supplementary materials, a more detailed description would enhance clarity.

2. I understand that the Authors use the assembly composed of a half-wave plate and a polarizing beam splitter to share the same light source between two different setups. However, this configuration, described also in the “didactic” section of the supplementary materials, leads to a situation where the control of the laser power is redundantly delegated to two different subsystems: this assembly and the Pockels cell. This would not be an issue in a general sense, but it conflicts with the nature of a simple and cost effective (so, almost minimalist) setup.

3. Some components appear in the CAD visualization of the setup but not in the cartoon visualization of the setup (eg. the beam block and the beam splitter in Fig. 1a and 1b). This discrepancy could cause confusion for readers, and harmonizing these figures would improve clarity.

4. On lines 213 and 214: non-SI units (feet) are used instead of SI units. These should be converted for consistency.

5. On lines 463 and following, point (6): it would be beneficial to mention that an IR LED is also needed to provide illumination for the wide-field configuration.

6. PLOS authors have the option to publish the peer review history of their article (what does this mean?). If published, this will include your full peer review and any attached files.

Reviewer #1: No

Reviewer #2: **Yes: **Giuseppe de Vito

---

## [Decision Letter · Decision Letter 1]

24 Jan 2025

TWINKLE: An open-source two-photon microscope for teaching and research

PONE-D-24-42429R1

Dear Dr. Schottdorf,

We’re pleased to inform you that your manuscript has been judged scientifically suitable for publication and will be formally accepted for publication once it meets all outstanding technical requirements.

Kind regards,

Luca Pesce, Ph.D.

Academic Editor

PLOS ONE

Additional Editor Comments (optional):

Reviewers' comments:

Reviewer's Responses to Questions

**Comments to the Author**

1. If the authors have adequately addressed your comments raised in a previous round of review and you feel that this manuscript is now acceptable for publication, you may indicate that here to bypass the “Comments to the Author” section, enter your conflict of interest statement in the “Confidential to Editor” section, and submit your "Accept" recommendation.

Reviewer #1: All comments have been addressed

Reviewer #2: All comments have been addressed

2. Is the manuscript technically sound, and do the data support the conclusions?

Reviewer #1: Yes

Reviewer #2: (No Response)

3. Has the statistical analysis been performed appropriately and rigorously? 

Reviewer #1: N/A

Reviewer #2: (No Response)

4. Have the authors made all data underlying the findings in their manuscript fully available?

Reviewer #1: Yes

Reviewer #2: (No Response)

5. Is the manuscript presented in an intelligible fashion and written in standard English?

Reviewer #1: Yes

Reviewer #2: (No Response)

6. Review Comments to the Author

Reviewer #1: The authors have properly addressed all the points raised in my comments, therefore I recommend the paper for publication.

Reviewer #2: (No Response)

7. PLOS authors have the option to publish the peer review history of their article (what does this mean?). If published, this will include your full peer review and any attached files.

Reviewer #1: No

Reviewer #2: **Yes: **Giuseppe de Vito

---

## [Editor Report · Acceptance letter]

30 Jan 2025

PONE-D-24-42429R1 

PLOS ONE

Dear Dr. Schottdorf, 

I'm pleased to inform you that your manuscript has been deemed suitable for publication in PLOS ONE. Congratulations! Your manuscript is now being handed over to our production team.

Kind regards, 

on behalf of

Dr. Luca Pesce 

Academic Editor

PLOS ONE